# Prevalence, Antibiotics Resistance and Plasmid Profiling of *Vibrio* spp. Isolated from Cultured Shrimp in Peninsular Malaysia

**DOI:** 10.3390/microorganisms10091851

**Published:** 2022-09-16

**Authors:** Wan Omar Haifa-Haryani, Md. Ali Amatul-Samahah, Mohamad Azzam-Sayuti, Yong Kit Chin, Mohd Zamri-Saad, I. Natrah, Mohammad Noor Azmai Amal, Woro Hastuti Satyantini, Md Yasin Ina-Salwany

**Affiliations:** 1Aquatic Animal Health and Therapeutics Laboratory, Institute of Bioscience, Universiti Putra Malaysia, Serdang 43400 UPM, Selangor, Malaysia; 2Department of Aquaculture, Faculty of Agriculture, Universiti Putra Malaysia, Serdang 43400 UPM, Selangor, Malaysia; 3Freshwater Fisheries Research Division, Fisheries Research Institute Glami Lemi, Jelebu 71650, Negeri Sembilan, Malaysia; 4Department of Veterinary Laboratory Diagnosis, Faculty of Veterinary Medicine, Universiti Putra Malaysia, Serdang 43400 UPM, Selangor, Malaysia; 5Department of Biology, Faculty of Science, Universiti Putra Malaysia, Serdang 43400 UPM, Selangor, Malaysia; 6Department of Aquaculture, Faculty of Fisheries and Marine, Universitas Airlangga, Surabaya 60115, Indonesia

**Keywords:** *Vibrio* spp., Peninsular Malaysia, antibiotics, MAR index, plasmid

## Abstract

*Vibrio* is the most common bacterium associated with diseases in crustaceans. Outbreaks of vibriosis pose a serious threat to shrimp production. Therefore, antibiotics are commonly used as preventative and therapeutic measures. Unfortunately, improper use of antibiotics leads to antibiotic resistance. Nevertheless, information on the occurrence of *Vibrio* spp. and antibiotic use in shrimp, particularly in Malaysia, is minimal. This study aimed to provide information on the occurrence of *Vibrio* spp., its status of antibiotic resistance and the plasmid profiles of *Vibrio* spp. isolated from cultured shrimp in Peninsular Malaysia. Shrimp were sampled from seven farms that were located in different geographical regions of Peninsular Malaysia. According to the observations, 85% of the shrimp were healthy, whereas 15% were unhealthy. Subsequently, 225 presumptive *Vibrio* isolates were subjected to biochemical tests and molecular detection using the *pyrH* gene. The isolates were also tested for antibiotic susceptibility against 16 antibiotics and were subjected to plasmid profiling. Eventually, 13 different *Vibrio* spp. were successfully isolated and characterized using the *pyrH* gene. They were the following: *V. parahaemolyticus* (55%), *V. communis* (9%)*, V. campbellii* (8%), *V. owensii* (7%), *V. rotiferianus* (5%), *Vibrio* spp. (4%), *V. alginolyticus* (3%), *V. brasiliensis* (2%), *V. natriegens* (2%), *V. xuii* (1%), *V. harveyi* (1%), *V. hepatarius* (0.4%) and *P. damselae* (3%). Antibiotic susceptibility profiles revealed that all isolates were resistant to penicillin G (100%), but susceptible to norfloxacin (96%). Furthermore, 16% of the isolates revealed MAR of less than 0.2, while 84% were greater than 0.2. A total of 125 isolates harbored plasmids with molecular weights between 1.0 and above 10 kb, detected among the resistant isolates. The resistant isolates were mediated by both chromosomal and plasmid factors. These findings support the use of surveillance data on the emerging patterns of antimicrobial-resistance and plasmid profiles of *Vibrio* spp. in shrimp farms. The findings from this study can be used to develop a better disease management strategy for shrimp farming.

## 1. Introduction

Shrimp is a popular seafood with high demand, especially in southeast Asia, providing a substantial contribution of USD 250 billion to the local, national, and regional economies [1,2]. China, India, Thailand, Indonesia, and Vietnam are the main contributors of white leg shrimp (*Penaeus vannamei*) and black tiger shrimp (*P. monodon*) with production values at USD 26.7 billion and USD 5.59 billion, respectively [1,2,3] Since global population is expected to reach over 9 billion people by 2030, shrimp aquaculture can play a major role in providing global food and nutritional security to people in both developed and developing countries, as well as sustaining the livelihood of the global population [2,4]. However, the pressure to intensify and expand shrimp aquaculture systems has exposed shrimp aquaculture to disease outbreaks, leading to huge economic losses of over USD 9 billion per year, which is approximately 15% of the value of global farmed fish and shellfish productions [2].

Antibiotics have played a critical role in reducing morbidity and mortality from bacterial diseases in aquaculture [5]. In fact, antibiotics and other drugs are commonly employed in shrimp culture for growth promotion, treatment, and disease prevention [6]. In addition, antibiotics such as oxytetracycline, tetracycline, quinolones, sulphonamides, enrofloxacin, norfloxacin, gentamicin and trimethoprim, are commonly used and permitted in shrimp farming [1,7,8]. However, antibiotics are heavily used in outbreak areas for immediate treatment, and the overuse of antibiotics in aquaculture eventually leads to antimicrobial resistance (AMR) in bacterial strains in fish and other aquatic species, including shrimp [9,10]. Eventually, several antibiotics are no longer effective in fighting bacterial infections [11]. Several alternatives have been considered to control disease outbreaks in shrimp farms, including strict biosecurity measures, green water systems, probiotics, and phage [12,13]. In fact, better farm management, including seed and stock selection, aeration, water treatment, and the application of non-harmful chemicals, such as organic acids and natural products in the diet, have been considered [14,15,16]. A good feed management system, including high-quality feed enriched with immunostimulants, as well as the addition of probiotics, aids in disease prevention [17]. Ignoring the biosecurity of shrimp hatcheries and farms allows outbreaks to spread.

Plasmid screening should be considered as an additional procedure in the monitoring programs to trace antibiotic resistance dissemination [18]. Plasmid plays a crucial role in the transmission of resistance genes, since it consists of genetic determinants of antibiotic resistance. In fact, there is a correlation between plasmid and antibiotic resistance among *Vibrio* spp. [19,20]. Therefore, plasmid curing is used to eliminate bacterial plasmid and determine antibiotic resistance mediation. Curing agents, such as acridine orange (AO), ethidium bromide (EB), and sodium dodecyl sulphate (SDS), are commonly used in plasmid curing [21]. Hence, this study aimed at providing important information regarding the prevalence, antibiotic resistance patterns, and plasmid profiling of *Vibrio* spp. isolated from cultured shrimp in Peninsular Malaysia.

## 2. Materials and Methods

### 2.1. Shrimp Sampling and In-Situ Examination

A total of 210 shrimp were randomly collected between March 2019 and March 2021 from seven farms (n = 30 shrimp/farm) that were located around Peninsular Malaysia (Table 1). They were *P. monodon* (n = 150) and *P. vannamei* (n = 60) having 30 days of culture. The shrimp samples included healthy and unhealthy shrimp and were randomly collected by using a lift net. During the collection, shrimp length and weight were measured and recorded. Then, the samples were immediately placed in ice and transferred to the laboratory within 3 to 4 h. Samples were washed thoroughly with sterile distilled water followed by dissection with sterile scissors. Following collection, the hepatopancreas and midgut were carefully examined. Healthy shrimp showed large and black hepatopancreases with full midguts, whereas unhealthy shrimp had small and pale hepatopancreases with empty midguts. Other clinical signs, such as body coloration, growth, size of the shrimp, and signs of poor feeding and swimming behaviour were observed [22,23,24].

### 2.2. Bacterial Isolation and Identification

Samples of hepatopancreases were homogenized in 1.0 mL of 0.9% sterile saline solution, and then serially diluted with sterile saline. One hundred µL of the suspensions were inoculated onto thiosulfate citrate bile salts sucrose (TCBS) agar (Oxoid, Hampshire, UK) and incubated for 24 h at 30 °C. The yellow and green colonies were then recorded and further sub-cultured onto TCBS to obtain a pure culture [22].

### 2.3. Biochemical Tests

All suspected *Vibrio* isolates were subjected to Gram staining (Becton Dickinson, Franklin Lakes, NJ, USA) before being subjected to a series of biochemical assays, which included Gram staining, triple sugar iron (TSI), oxidase, catalase, O-nitrophenyl beta-D-galactosidase (ONPG), sulphur production, indole, motility (SIM) agar and lysine decarboxylase (LDC) (Oxoid, Hampshire, UK) [25,26].

### 2.4. Bacterial Identification Using pyrH Gene

Genomic DNA of the suspected *Vibrio* isolates was extracted using the DNA Purification Kit (Promega, Madison, WI, USA) and stored at −20 °C until use. Identification of the *Vibrio* isolates was verified based on the partial sequencing of *pyrH* that produced a PCR product size of around 500 bp. The *pyrH* forward primer of 5′-GAT CGT ATG GCT CAA GAA G-3′ and *pyrH* reverse primer of 5′-TAG GCA TTT TGT GGT CAC G-3′ were used [27]. The PCR cycling condition included an initial denaturation at 94 °C for 3 min, followed by 30 cycles of denaturation at 94 °C for 1 min, annealing at 55.3 °C for 2 min 15 s, extension at 72 °C for 1 min 15 s, and a final denaturation at 94 °C for 1 min 15 s. The PCR mixtures containing 2.5 μL of the DNA template (10–100 ng) were mixed with 12.5 μL of premixed Reddiant PCR MasterMix (First Base, Kuala Lumpur, Malaysia) comprised of 60 U/mL of Taq DNA polymerase, 3 mM MgCl_2_, 400 µM dNTP mix and 10 µM each of forward and reverse primers, in a total volume of 25 μL of PCR mixture. Then, the PCR amplifications were carried out using a PCR thermocycler (Bio-Rad, Hercules, CA, USA). The amplified products were examined using 1% agarose gel electrophoresis that had been pre-added with FloroSafe DNA stain (First Base Laboratories) dye and sequencing was performed (Apical Scientific Sdn. Bhd., Kuala Lumpur, Malaysia).

### 2.5. Sequence Editing and Analysis

The resulting DNA sequences were trimmed of low quality at the frond and end of the sequence by using ChromasPro 2.1.8 software (Technelysium Pty Ltd., South Brisbane, Queensland, Australia). The trimmed sequences were then blasted in the GenBank database using the nucleotide Basic Local Alignment Search Tool (BLASTn) (https://blast.ncbi.nlm.nih.gov/Blast.cgi (accessed on 1 April 2019)) for preliminary identification.

### 2.6. Phylogenetic Analysis

Phylogenetic analysis was conducted using MEGA version 7.0 [28]. After the addition of *pyrH* genes obtained from 36 *Vibrio* spp. reference genomes (Table A1), the database of *pyrH* sequences was aligned using ClustalW and a curated database of the sequences was used for subsequent analyses. A neighbor joining phylogenetic analysis was conducted with the Kimura-2-parameter model with 1000 bootstrap replicates [29]. The reported sequences were deposited in the GenBank nucleotide sequence databases (accession numbers OP198216–OP198433). (Appendix A).

### 2.7. Antibiotic Susceptibility Test

Antibiotic susceptibility of the *Vibrio* isolates was determined using the disc diffusion method [30]. Sixteen antibiotics (Thermo Fisher Scientific, Waltham, MA, USA) were used, which included the following: tetracycline 30 μg (TET), ampicillin 10 μg (AMP), gentamicin 10 μg (CN), sulfomethiozole-trimethoprim 25 μg (SXT), erythromycin 15 μg (E), chloramphenicol 30 μg (C), norfloxacin 10 μg (NOR), penicillin G 10U (P), cefepime 30 μg (FEP), cefotaxime 30 μg (CTX), ceftazidime 30 μg (CAZ), cephalothin 30 μg (KF), kanamycin 30 μg (K), ciprofloxacin 5 μg (CIP), nitrofurantoin 300 μg (F) and vancomycin 30 μg (VA). Following incubation for 24 h at 30 °C, the isolates were then inoculated in sterile saline water to achieve a bacterial concentration of 1.5 × 10^8^ CFU/mL, which was equivalent to 0.5 MacFarland standard. The broth was evenly swabbed onto Mueller Hinton (MH) agar (Oxoid, Hampshire, UK) supplemented with 1.5% of sodium chloride (NaCl) [31]. Antibiotic discs were aseptically placed on the agar and incubated at 35 °C for 18 h before the inhibition zones were measured. Antibiotic susceptibilities, categorized as resistant (R), intermediate (I) or susceptible (S), were assigned using criteria outlined in the Clinical Laboratory Standards Institute (version M45-A2) [32,33]. The multiple antibiotic resistance (MAR) index was calculated using the MAR index value: A/B = MAR index, where A was the number of antibiotics to which the isolate was resistant, and B was the total number of antibiotics used in this study [34]. A MAR index value of less than 0.2 indicated that the isolates were from low-risk sources of contamination. In contrast, a MAR index of greater than 0.2 indicated that the isolates were from high-risk sources of contamination [35].

### 2.8. Plasmid Extraction

Approximately 2.5 mL of bacterial culture from Tryptic Soy Broth (TSB) (Oxoid, Hampshire, UK), supplemented with 1.5% NaCl, was centrifuged at 12,000× *g* for 3 min. The *Vibrio* isolate was purified using a GeneJet Plasmid Purification kit (Thermo Fisher Scientific, Waltham, MA, USA). The plasmid-containing supernatant was stored at −20 °C. The presence of plasmid was detected using agarose gel (1% *w*/*v*) electrophoresis (Bio-Rad).

### 2.9. Plasmid Curing

*Vibrio* isolates, that harbored plasmid, were treated with acridine orange (AO) (Thermo Fisher Scientific, Waltham, MA, USA) following modifications of the method by Letchumanan et al. [19]. The isolates were grown on TSB supplemented with 1.5% of NaCl, at 30 °C for 18 h under constant agitation. After incubation, 200 µL of the aliquots were added into the tubes containing Luria Bertani (LB) broth (Oxoid, Hampshire, UK), supplemented with 1.5% of NaCl (control), and into the tubes containing LB broth, supplemented with 1.5% of NaCl and AO. The isolates were incubated at 30°C for 18 h [36]. The presence of plasmid was detected using agarose gel electrophoresis (1 % *w*/*v*) after treatment with AO. In addition, the antibiotic susceptibility test was repeated, as stated earlier, to confirm changes in resistance profiles.

## 3. Results

### 3.1. Clinical Examination

Most of the sampled shrimp were healthy (85%), while 15% were unhealthy shrimp. Based on the observations, the hepatopancreas of healthy shrimp was large and black, while the midgut was full (Figure 1a,c). On the other hand, the hepatopancreas of unhealthy shrimp appeared small and pale, while the midgut was empty (Figure 1b,d). The body coloration between healthy and unhealthy seemed to be different, in that unhealthy shrimp had a pale body color (Figure 1b) compared to healthy shrimp (Figure 1a). In fact, the sizes of the unhealthy shrimp were smaller compared to healthy shrimp. The in-situ examination showed that the hepatopancreas and midgut were associated with signs of vibriosis. However, additional tests, in terms of molecular identification and histological examination, are required for verification.

### 3.2. Isolation and Identification of Vibrio spp.

A total of 225 suspected *Vibrio* isolates were obtained from the 210 sampled shrimp. They were identified by color, shape, and size of the culture on TCBS agar. Among the 225 isolates, 149 (66%) isolates produced green, round, and medium-sized (2–5 mm) colonies, while 76 (34%) isolates produced yellow, round, and medium-sized colonies (Table 2).

### 3.3. Biochemical Characterization of Vibrio spp.

All 225 isolates (100%) were Gram negative, positive to catalase and oxidase tests, motile but not able to produce sulfide gas. An indole test showed that 95% of the isolates were positive, while 80% of the isolates tested positive for lysine dehydrogenase (LDC). None of the isolates produced gas and hydrogen sulfide. However, 61% of the isolates produced red (alkaline) coloration in the slants and yellow (acid) in the butts. They fermented the glucose. A total of 39% appeared yellow in both the slants and butts (Table 3).

### 3.4. Prevalence of Vibrio spp.

A total of 225 suspected *Vibrio* spp. were isolated based on the green and yellow colonies appearing on thiosulfate citrate bile salt sucrose (TCBS) agar. They were also *pyrH* positive producing a single band of 500 bp. Among them, 22% of the isolates were from Banting, Selangor, 16% were from Ayer Hitam, Kedah, 16% were from Marang, Terengganu, 13% were from Manjung, Perak, 12% were from Merlimau, Melaka, 11% were from Sg. Besar, Selangor and 11% were from Mersing, Johor (Figure 2). As shown in Figure 2, *V. parahaemolyticus* was found in most of the sampling sites, including Banting in Selangor, followed by Sg. Besar in Selangor, Merlimau in Melaka, Mersing in Johor, Marang in Terengganu and Manjung in Perak. While *V. owensii* found in Ayer Hitam in Kedah, Marang in Terengganu, Manjung in Perak and Banting in Selangor. *V. campbellii* were found in Sg. Besar in Selangor, Merlimau in Melaka and Manjung in Perak. Moreover, *V. campbellii* was the dominant species in Manjung, Perak and Sg. Besar, Selangor. Meanwhile, *V. rotiferanius* was found predominantly in Ayer Hitam, Kedah. It was also found in Sg. Besar in Selangor, Merlimau in Melaka and Manjung in Perak. *V. alginolyticus* were found in Mersing, Johor and Ayer Hitam, Kedah. *V. communis* was mostly found in Ayer Hitam Kedah and was also detected in Sg. Besar, Selangor. *P.damselae* was found in Sg Besar and Banting, Selangor. It was predominantly found in Merlimau, Melaka. Other species, including *V. xuii* and *V. harveyi*, were only found in Manjung, Perak. *V. brasiliensis* was found in Ayer Hitam, Kedah, while *V. hepatarius* and *V. natriegens* were only found in Banting, Selangor.

### 3.5. Phylogenetic Analyses of Vibrio spp.

The 225 *Vibrio* strains were successfully isolated and identified from 210 cultured shrimp. Phylogenetic analysis of *pyrH* sequences revealed that 96% of the isolates were clustered into 12 distinct species, while the remaining 4% were clustered into *Vibrio* spp. (Figure 3). The strains were categorized into three clades, including Harveyi, Nereis and Orientalis. The Harveyi clade consisted of eight species, which were *V. parahaemolyticus*, *V. owensii*, *V. communis*, *V. rotiferianus*, *V. campbellii*, *V. alginolyticus*, *V. natriegens*, and *V. harveyi**. V. xuii* belonged to the Nereis clade. On the other hand, the Orientalis clade consisted of *V. brasiliensis* and *V. hepatarius*. There were a total of 55% isolates clustered into *V. parahaemolyticus*, followed by 7% of *V. owensii*, 9% of *V. communis*, 5% of *V. rotiferianus*, 8% of *V. campbellii*, 3% of *V. alginolyticus*, 2% of *V. brasiliensis*, 1% of *V. xuii* and *V. harveyi*, 2% of *V. natriegens* and 0.4% of *V. hepatarius* (Figure 3). In this study, all the *Vibrio* spp. were above 95% threshold to distinguish between *Vibrio* species, as proposed by Sawabe et al. [37].

### 3.6. Antibiotics Susceptibility Test

The findings revealed that all the *Vibrio* isolates (13/13) were found to be resistant to vancomycin (VA), including *V. parahaemolyticus*, *V. campbellii*, *V.rotiferianus*, *V. owensii*, *V. alginolyticus*, *V. xuii*, *V. harveyi*, *V. natriegens*, *V. hepatarius*, *V. communis*, *V. brasiliensis*, *Vibrio* spp. and *P. damselae.* The findings showed that eight out of thirteen species were highly resistant (100%) to vancomycin (VA). The same applied to penicillin G (P), which all of the *Vibrio* isolates (13/13) were found to be resistant to. However, seven out of thirteen (7/13) species were highly resistant (100%) to penicillin G. The results also indicated that all *Vibrio* spp. were resistant to at least one of the 16 antibiotics tested. When compared to other *Vibrio* spp. isolates, *V. parahaemolyticus* was found to be resistant to all the tested antibiotics (16/16). Indeed, *V. parahaemolyticus* was discovered to be the only species that was resistant to norfloxacin (NOR) (1%). The findings also indicated that seven out of thirteen (7/13) of the *Vibrio* isolates, over 50%, were resistant toward the tested antibiotics, including *V. parahaemolyticus*, *V. campbellii*, *V. rotiferianus*, *V. owensii*, *V. alginolyticus*, *V. natriegens* and *V. communis.* (Table 4).

### 3.7. Multiple Antibiotic Resistance (MAR) Index

The MAR index ranged between 0.06 and 0.75 (Table 5). The average of MAR index was 0.41, and 16% (n = 36) had a MAR index of <0.2, whereas the remaining 84% (n = 189) showed a MAR index of >0.2. In fact, 0.4% had a MAR index of 0.06, indicating that they were antibiotic resistant to at least one type of antibiotic. On the other hand, 0.4% of the isolates showed a MAR index of 0.75, indicating a resistance to 12 antibiotics. Approximately 95% of the isolates were multidrug resistant (MDR), meaning that they were resistant to three or more antibiotics, with a MAR index between 0.19 and 0.75 (Table 5). The most common MAR index was 0.38, which was found in 22% of the *Vibrio* isolates, indicating that they were resistant to six different antibiotics. Other MAR indices were 0.13 (4%), 0.19 (12%), 0.25 (15%), 0.31 (16%), 0.44 (13%), 0.50 (6%), 0.56 (6%), 0.63 (2%) and 0.69 (3%).

### 3.8. Plasmid Profiling

Among the 225 *Vibrio* isolates tested, 125 (55.6%) isolates harbored plasmid with molecular weight from 1.0 to above 10.0 kb. While 100 (44.4%) of the isolates did not harbored any of the plasmid. The *Vibrio* spp. that were found to contain plasmid including *V. parahaemolyticus* (80%), *V. communis* (40%), *V. campbellii* (44%), *V. owensii* (60%), *V. rotiferanius* (42%), *V. natriegens* (40%), *V. brasiliensis* (50%), *V. alginolyticus* (71%), *Vibrio* spp. (38%) and *P. damselae* (43%). However, none of the plasmid were found in *V. harveyi*, *V. xuii* and *V. hepatarius* (Table 6). Most of the isolates, that harbored plasmid, were resistant to ampicillin (AMP) (84%) (Table 6). The percentages of plasmids discovered in the resistance isolates are shown in Table 7. There were 57.6% that had one plasmid, 15.2% had two plasmids, 18.4% had three plasmids, 3.2% had four and six plasmids and 2.4% had five plasmids. The plasmids were categorized into twenty-three (23) different profiles as follows: 4 (3.2%) of the isolates presented profiles 1 and 23; 1 (0.8%) of the isolates presented profiles 2, 5, 7, 9, 10, 11,13, 15 and 21; 2 (1.6%) of the isolates presented profiles 3, 6, 8, 14, 16, 19, 20 and 22; 65 (52.0%) of the isolates presented profile 4; 9 (7.2%) of the isolates presented profile 12; and 5 (4.0%) of the isolates presented profile 17 (Table 7). From 23 of the plasmid profiles, the profile that formed the largest group was plasmid profile 4 that consisted of one plasmid of above 10.0 kb size (Table 7).

### 3.9. Plasmid Curing

Figure 4 shows the number of resistant isolates before and after plasmid curing. The number of resistant isolates were reduced after the curing. Generally, the number of resistant isolates changed to intermediate and susceptible after the curing process with acridine orange. According to Figure 4a, it was shown that before the curing process, 125 of the isolates were resistant to ampicillin (AMP). However, after curing, 110 of the isolates remained resistant to ampicillin (AMP), while 15 of the isolates changed to either intermediate or susceptible, 4 to intermediate and 11 to susceptible (Figure 4b). Similarly, from 125 of the isolates, 123 and 124 of the isolates remained resistant to vancomycin (VA) and penicillin G (P), respectively, after the curing process. On the other hand, the number of isolates that were resistant to cephalothin (KF) drastically dropped from 102 to 57 isolates (Figure 4a), 28 changing to intermediate and 17 to susceptible (Figure 4b). There was also reduction in resistance of isolates to cefotaxime (CTX), from 93 to 55 isolates (Figure 4a), 19 of the isolates changing to intermediate and susceptible, respectively (Figure 4b). Resistance of the isolates against kanamycin (K) decreased from 76 to 42 isolates (Figure 4a), 33 changing to intermediate and 1 to susceptible (Figure 4b). In addition, a similar pattern of reduction was observed for isolates’ resistance against nitrofurantoin (F), from 58 before curing to 31 after curing (Figure 4a), 21 changing to intermediate and 6 to susceptible (Figure 4b). The findings also revealed a dramatic decrease in isolates resistant to ceftazidime (CAZ) and erythromycin (E), from 37 to 7 and 26 to 6, respectively (Figure 4a), with 11 and 19 changing to intermediate and susceptible regarding ceftazidime (CAZ) and 12 and 8 changing to intermediate and susceptible against erythromycin (E) (Figure 4b). All of the isolates resistant to cipfloxacin (CIP) changed to intermediate (7) and susceptible (6) (Figure 4a,b). Based on the gentamicin (CN) results, the number of resistant isolates dropped from 9 to 3 isolates (Figure 4a), 3 changing to intermediate and 3 to susceptible (Figure 4b). The number of the isolates resistant to cefepime (FEP) dropped from 8 to 6 isolates (Figure 4a), 2 changing to susceptible after the curing process. The number of isolates resistant to chloramphenicol (C) dropped from 12 to 11 isolates (Figure 4a), one of the isolates changing to intermediate (Figure 4b). For norfloxacin (NOR), the resistance of the isolates changed to susceptible after the curing process (Figure 4a,b). There was no change in resistance to sulfomethiozole-trimethoprim (SXT) and tetracycline (TET) after plasmid curing, indicating that the resistance isolates were chromosomally mediated (Figure 4a). The result after plasmid curing revealed that when the antibiotic resistance profile was affected, it indicated that the resistance isolate was plasmid mediated, whereas when the antibiotic resistance profile was unaffected, it indicated that the resistance isolate was chromosomal mediated [19].

## 4. Discussion

*Vibrio* spp. are a group of bacteria naturally found in freshwater, estuaries and marine environments [38]. It is commonly known that *Vibrio* spp. are responsible for numerous human diseases attributed to the natural microbiota of aquatic environments and seafood [39]. In humans, *Vibrio* spp. are known to cause gastroenteritis, cholera, and septicemia [40]. The human pathogenic *Vibrio* spp. of clinical relevance include the following: *V. parahaemolyticus*, *V. alginolyticus*, *V. cholerae*, *V. vulnificus*, *V. tubiashi* and *V. fluvialis* [41]. In addition, *V. parahaemolyticus*, *V. vulnificus*, and *V. mimicus* are foodborne pathogens [42]. Meanwhile, the common *Vibrio* spp. associated with the shrimp diseases are *V. harveyi*, *V. parahaemolyticus*, *V. alginolyticus*, *V. anguillarum*, *V. vulnificus* and *V. splendidus*. *V. harveyi* is associated with luminescent vibriosis in shrimps, particularly in *P. vannamei* and *P. monodon*. *V. parahaemolyticus* can cause human illness and is frequently associated with food-borne gastroenteritis or diarrhoea [43].

Identification based on biochemical keys has been proposed for *Vibrio* spp. and is an excellent means to obtain a large number of reference and environmental *Vibrio* strains [34]. However, conventional phenotyping and biochemical identification techniques are poorly adapted to *Vibrio* strains isolated from seafood and aquatic environments [44]. The presence of both false positive and false negative results in all the biochemical identification methods leads to difficulties identifying the *Vibrio* spp. [45]. Therefore, PCR assays are required for their identification and detection [46].

Phylogenetic analyses were carried out using the neighbor-joining method to confirm taxonomic position in the genus *Vibrio* [47]. Generally, the 16S rRNA gene is the most popular molecular marker for identifying and classifying isolated pure cultures and estimating bacterial density in environmental samples through metagenomic assay [48]. However, the 16S rRNA gene has low discriminatory power, leading to the misidentification of *Vibrio* spp. compared to other tested genes [37]. In addition, *Vibrio* spp. share more than 97% similarity with the 16S rRNA resulting in difficulties in differentiating closely related species [49]. Due to this limitation, other housekeeping genes are used as phylogenetic markers to determine the diversity of bacterial species. In this study, phylogenetic analysis using the *pyrH* gene showed that *Vibrio* spp. isolated from shrimp belonged to three clades: Harveyi, Nereis and Orientalis. The Harveyi clade consists of eight species, which are *V. parahaemolyticus*, *V. owensii*, *V. communis*, *V. rotiferianus*, *V. campbellii*, *V. alginolyticus*, *V. natriegens*, and *V. harveyi. V. xuii* belongs to the Nereis clade. Finally, the Orientalis clade consists of *V. brasiliensis* and *V. hepatarius* [37]. The *pyrH* gene is a housekeeping gene that encodes uridine monophosphate kinase (UMP kinase), which participates in the pyrimidine biosynthesis catalyzing the conversion of UMP into UDP [50]. According to Chimetto et al. [51], the *pyrH* gene could effectively distinguish the species level of *Vibrio*, including *V. communis*, which currently is categorized as *Vibrio* spp. In fact, *pyrH* were also among the genes that gave the highest resolution in distinctively differentiating *V. harveyi* and *V. campbellii* [52]. In addition, it was also among the genes that gave the highest resolution in distinctively differentiating *V. harveyi* and *V. campbellii* [52]. Thus, *pyrH* gene was chosen, since it was one of the molecular markers suitable for identifying the closely related *Vibrio* spp. in this investigation [53].

A study performed by Thompson et al. [54] compared the species resolution level with three different housekeeping genes, *rpoA*, *recA* and *pyrH*. The findings revealed that the *pyrH* gene was a good predictor of *Vibrio* and a good discriminatory target at the species level. Moreover, they showed stability of this locus, due to high proportions of synonymous mutations leading to the conservation of the amino acid sequence. Furthermore, until other genome-based techniques are proposed, *pyrH* is the most powerful method for distinguishing *Vibrio* spp. in biodiversity, population genetics, and evolution studies. It also helps to eliminate the misidentification of *Vibrio* ancestry clades [37]. In molecular phylogenetics, the use of a minimum gene set in molecular phylogenetics is critical for reducing time and expense, while also improving the accuracy of results. This is especially important when identifying species and understanding population structure and evolution in a large bacterial taxon like the Vibrionaceae family, with over 140 species [37].

In this study, we successfully isolated 225 *Vibrio* spp. from the hepatopancreases of cultured shrimp. They were *V. parahaemolyticus*, *V. owensii*, *Vibrio* spp., *V. communis*, *V. rotiferianus*, *V. campbellii*, *V. alginolyticus*, *V. brasiliensis*, *V. xuii*, *V. harveyi*, *V. natriegens*, *V. hepatarius* and *P. damselae*. *Vibrio parahaemolyticus* was the most prevalent species, isolated from six sampling locations. In fact, *V. parahaemolyticus* is often isolated from seafood, particularly shellfish or bivalve mollusks, worldwide [55]. In Croatia, 9.4% of sea fish, shrimp, and bivalve mollusks carry *V. parahaemolyticus* [56]. Similarly, Letchumanan et al. [57] revealed that *V. parahaemolyticus* could easily be isolated from retail shrimp in Malaysia. *Vibrio parahaemolyticus* was found to be dominant in shrimp in Ecuador (81%), Sri Lanka (98%), and Egypt (18%) [58,59,60]. Other species, such as *V. campbellii* and *V. harveyi*, were common species detectable in tropical marine regions and are among the most important bacterial pathogens of many commercially farmed marine invertebrate and vertebrate species in many Asian countries [61,62]. However, many previous studies have demonstrated the predominance of *V. alginolyticus* in shrimp or seafood samples [63]. According to Kriem et al. [44], the most common *Vibrio* spp. in shrimp in Morocco was *V. alginolyticus*. Similarly, Baffone et al. [64] reported the predominance of *V. alginolyticus* among fresh seafood products, followed by *V. parahaemolyticus* and *V. cholerae*. Unfortunately, contrary to our findings, *V. alginolyticus* was scarce and could only be found in two sampling locations in Mersing, Johor and Ayer Hitam, Kedah. The source of samples, methods of identification, study area, season, salinity, and temperature during storage or even transportation may influence the variation of *Vibrio* spp. prevalence in seafood [65,66]. In addition, temperature, pH, salinity, and nutrient levels present in the water column can ultimately affect the abundance of *Vibrio* [67].

Antibiotic resistance can be transmitted through sequential mutations in chromosomal genes or by acquiring genetic elements, such as plasmids, bacteriophages, or transposons [68]. This study demonstrated that 100% of the isolates were resistant to penicillin G, followed by vancomycin (98%) and ampicillin (84%). *Vibrio* resistance to ampicillin has been reported since 1978 with ranges between 40% and 90%. The ampicillin-resistant pattern could be related to the abuse of first-generation antibiotics in the environment, which reduced ampicillin susceptibility and efficiency in treating *Vibrio* infection [69]. Similarly, penicillin-resistant *Vibrio* has already been reported in different penaeid culture regions [70,71]. Srinivasan and Ramasamy [72] reported 100% penicillin G resistance in India, while Albuquerque et al. [73] also revealed high resistance to penicillin G. This study also reported that the isolates were highly resistant to vancomycin, which was consistent with Noorlis et al. [28]. Chloramphenicol (C) and norfloxacin (NOR) were the antibiotics to which *Vibrio* spp. were least resistant. The study showed that resistance to chloramphenicol (C) was only found in *V. parahaemolyticus* (9%) and *V. communis* (5%), while resistance to norfloxacin was only found in *V. parahaemolyticus* (1%). The result regarding chloramphenicol was expected, since it was banned due to the risks of human exposure to its residues in food products, which prevents fish farmers from using it [69]. Previous research by Ottaviani et al. [74], Sahilah et al. [75], and Sudha et al. [69] also supported our findings. Hence, regular monitoring on the usage of antibiotics is required to avoid the emergence of new resistant strains.

The Multiple Antibiotic Resistance (MAR) index, which ranges from 0 to 1.0, is a valuable measure for determining health risks. The MAR index value (0.20) is differentiated between low and high risks. A MAR value greater than 0.20 indicates that the samples have an increased risk of source contamination, while a MAR less than 0.2 indicates that the samples have a low risk of source contamination [35]. Our findings revealed a significant frequency of MAR index >0.2 (84%) in the sampling area compared to MAR index <0.2 (16%), thereby indicating that the aquatic environment in the sampling area may be affected and contaminated with antibiotics from human, animal, and fish consumer sources. In this study, although the farmers claimed that they did not use any antibiotics on their farms, the findings revealed that the occurrence of resistance to multiple antibiotics was very high. This might have been due to the sampling site being located in an area near the city, agricultural or industrial sectors. Hence, the resistance of bacteria to antibiotics might have travelled through water from nearby farms to the shrimp ponds. In fact, antibiotics from animal feeds or medications are absorbed into the sediment causing bacterial selection in the nearby environment [76]. Another possibility was that the presence of antibiotic-resistant bacteria in shrimp could be from post larvae, and the associated variety of drugs used in shrimp hatcheries [40]. The use of these drugs causes resistance to certain antimicrobials during post larvae rearing in the hatchery, which persists in the shrimp gut after transfer to the grow out ponds [77]. Furthermore, geographical location differences may influence the resistance level variance depending on sample collection [18].

The findings revealed that more than 55% of the isolates harbored between one to six DNA bands of plasmid, similar to those reported by Zanetti et al. [78] and You et al. [79]. Plasmid profile determination has been found to be very useful in epidemiological studies, and diagnosis and elucidation of mechanisms of drug resistance [80]. Plasmids are extra-chromosomal materials that allow the movement of genetic materials, including antimicrobial-resistant genes, between bacterial species and genera, through gene exchange processes; hence, increasing antibiotic resistance [81]. A study by Devi et al. [30] revealed that there was no correlation between the resistance of *Vibrio* to antibiotics and the presence of plasmid. Even though an isolate does not exhibit plasmids, the isolate can still show antibiotic resistance [82]. This statement was also supported by Zulkifli et al. [83], whose findings revealed that 53% of the isolates did not harbor any plasmid. However, they also showed multiple antibiotic resistance patterns to high number of antibiotics, which indicated that resistance to most of these antibiotics was of chromosomal origin, or on mobile genetic elements, that might help in the dissemination of resistant genes to other bacteria of human clinical significance. Nevertheless, most *Vibrio* isolates in this study lost their resistance to antibiotics following plasmid curing, even though some remained resistant. This indicates that the antibiotic resistance genes in *Vibrio* spp. isolated from cultured shrimp in this study were both chromosomal and plasmid mediated [36,73]. Although antimicrobial resistance in *Vibrio* spp. is typically acquired through plasmid transfer or antibiotic exposure, the role of plasmids in multiple antimicrobial resistances should be investigated further [74].

## 5. Conclusions

The findings thoroughly analyzed the occurrence, antibiotic resistance profiles, and plasmid profiling of *Vibrio* spp. isolated from cultured shrimp in Malaysia. Hence, antimicrobial resistance surveillance and drug usage monitoring in aquaculture should be encouraged to improve antibiotic management for public health and food safety in the industry. Furthermore, better knowledge of the molecular basis of resistance acquisition and transmission can aid in the development of new strategies to combat vibriosis for sustainable shrimp production.

## Figures and Tables

**Figure 1 microorganisms-10-01851-f001:**
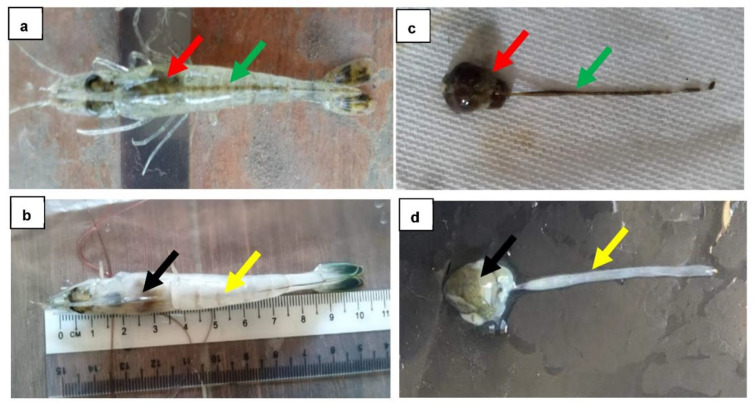
Gross signs between healthy and unhealthy shrimp. (**a**,**c**); healthy shrimp had a large pigmented hepatopancreas (red arrow) and full midgut (green arrow). (**b**,**d**); unhealthy shrimp had a pale, atrophied hepatopancreas (black arrow) and an empty midgut (yellow arrow). (**c**,**d**) are hepatopancreases individually dissected from the shrimp.

**Figure 2 microorganisms-10-01851-f002:**
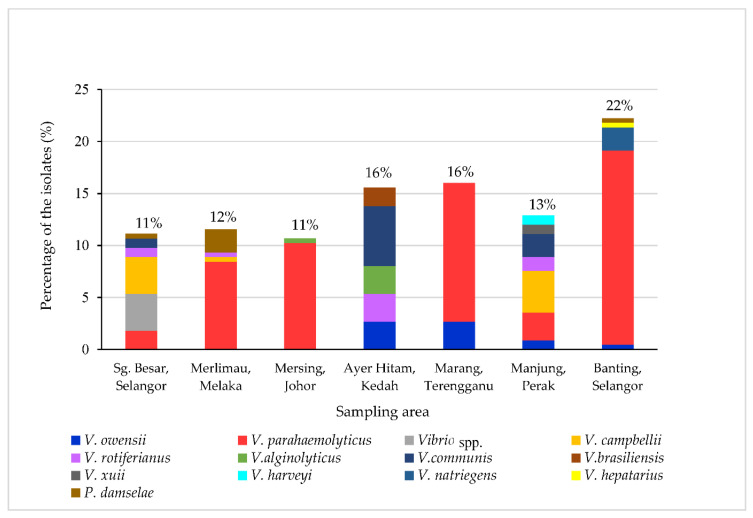
The prevalence of *Vibrio* spp. from seven farms in Peninsular Malaysia.

**Figure 3 microorganisms-10-01851-f003:**
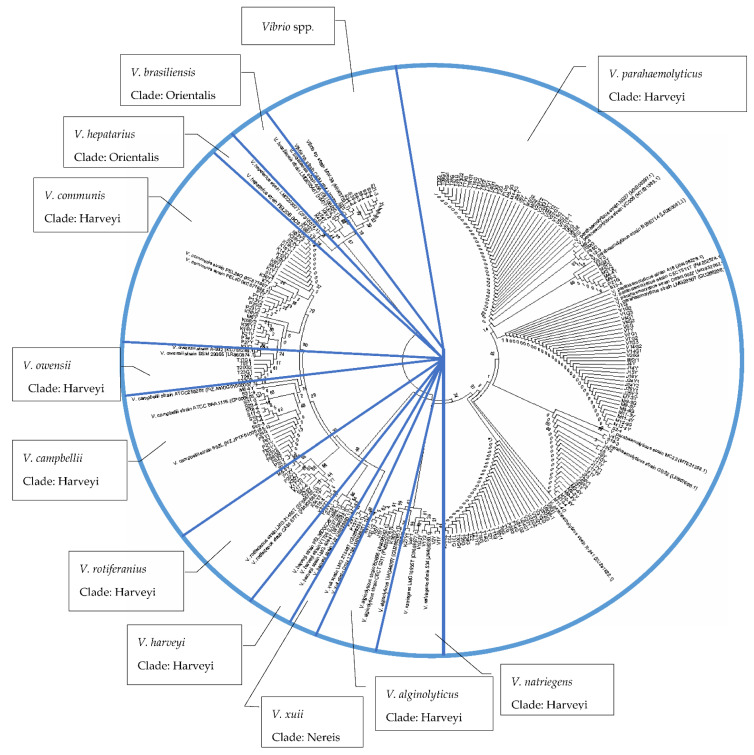
Neighbor-joining phylogeny (Kimura 2-parameter model) of concatenated partial *pyrH* gene sequences from *Vibrio* spp. isolated from cultured shrimp. Bootstrap values represent 1000 replications. Reference sequences were acquired from the NCBI GenBank.

**Figure 4 microorganisms-10-01851-f004:**
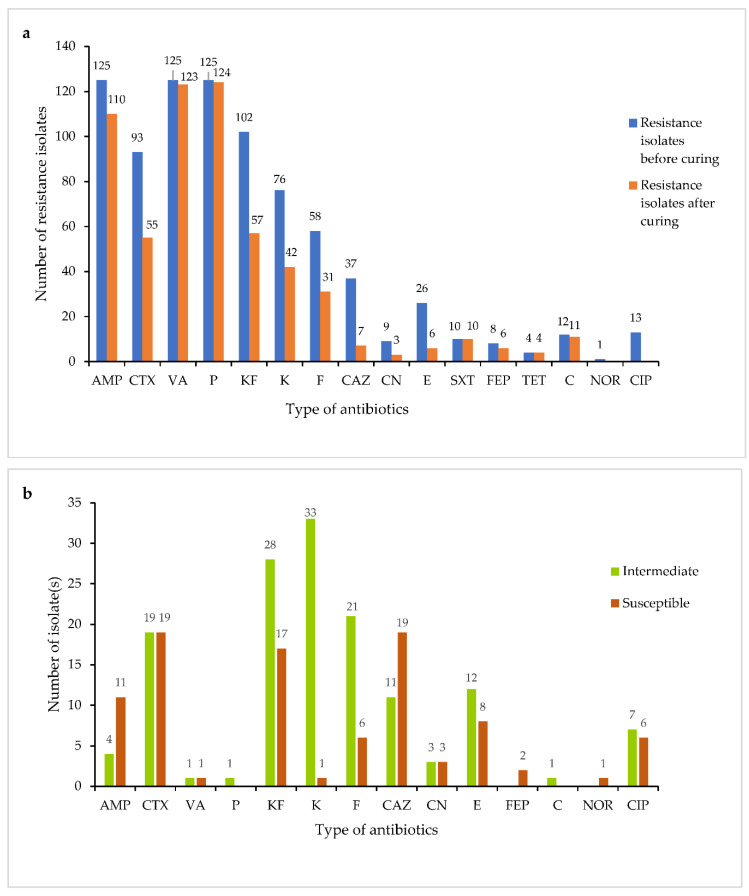
(**a**) Antibiotic resistance profiles of *Vibrio* spp. before and after plasmid curing. The Y-axis represents the number of isolates resistant to the tested antibiotics, while the X-axis represents types of antibiotics (**b**) Number of isolates that changed to intermediate and susceptible from resistance after plasmid curing. AMP: Ampicillin, CTX: Cefotaxime, VA: Vancomycin, P: Penicillin G, KF: Cephalothin, K: Kanamycin, CAZ: Ceftazidime, F: Nitrofurantoin, CN: Gentamicin, E: Erythromycin, SXT: Sulfomethiozole-trimethoprim, FEP: Cefepime, TET: Tetracycline, C: Chloramphenicol, NOR: Norfloxacin, CIP: Ciprofloxacin.

**Table 1 microorganisms-10-01851-t001:** Shrimp sampling data from seven shrimp farms in different geographical regions of Peninsular Malaysia.

Sampling Site	State	Region	Types of Ponds	Number of Shrimp	Shrimp Species	Date	Coordinate
Sg. Besar	Selangor	Central	Earthen	30	*P. monodon*	March 2019	3.7726 N 100.9666 E
Merlimau	Melaka	West	Earthen	30	*P. monodon*	April 2019	2.0631 N 102.2910 E
Mersing	Johor	South	Earthen	30	*P. monodon*	January 2020	2.2941 N 103.4902 E
Ayer Hitam	Kedah	North	HDPE lined earth	30	*P. monodon*	March 2020	6.2438 N 100.2220 E
Marang	Terengganu	East	HDPE lined earth	30	*P. vannamei*	December 2020	5.1846 N 103.1991 E
Manjung	Perak	North	Earthen	30	*P. vannamei*	January 2021	4.3722 N 100.6089 E
Banting	Selangor	Central	Earthen	30	*P. monodon*	March 2021	2.8222 N 101.4179 E

Abbreviation: HDPE: High Density Polyethylene.

**Table 2 microorganisms-10-01851-t002:** Total number of *Vibrio* spp. from the cultured shrimp based on the color of the colony on TCBS.

Sampling Site	State	Total Number of Isolates, n (%)	Number of Isolates on TCBS, n (%)	
			Green Colony	Yellow Colony
Sg. Besar	Selangor	25 (11)	13 (6)	12 (5)
Banting	Selangor	50 (22)	43 (19)	7 (3)
Merlimau	Melaka	26 (12)	25 (11)	1 (0.4)
Mersing	Johor	24 (11)	23 (10)	1 (0.4)
Ayer Hitam	Kedah	35 (16)	0 (0)	35 (16)
Marang	Terengganu	36 (16)	30 (13)	6 (3)
Manjung	Perak	29 (13)	15 (7)	14 (6)
Total		225 (100)	149 (66)	76 (34)

Abbreviation: TCBS, thiosulphate citrate bile salt sucrose.

**Table 3 microorganisms-10-01851-t003:** Biochemical properties of *Vibrio* spp. isolated from cultured shrimp in Peninsular Malaysia.

Species	Number of Isolates	TCBS	Gram Staining	Oxidase	Catalase	ONPG	SIM			TSI			LDC
							Sulphide Gas Production	Indole Production	Motility	A/A, K/A	Gas Production	H_2_S Production	
*V. owensii*	15	Y	negative	+	+	+	-	+	+	A/A	-	-	+
*V. parahaemolyticus*	124	G	negative	+	+	+	-	+	+	K/A	-	-	+
*Vibrio* spp.	8	Y	negative	+	+	-	-	+	+	A/A	-	-	-
*V. campbellii*	18	G	negative	+	+	-	-	+	+	A/A	-	-	-
*V. rotiferianus*	12	Y	negative	+	+	-	-	+	+	K/A	-	-	+
*V. alginolyticus*	7	Y	negative	+	+	-	-	+	+	A/A	-	-	+
*V. communis*	20	Y	negative	+	+	-	-	+	+	A/A	-	-	+
*V. brasiliensis*	4	Y	negative	+	+	+	-	+	+	A/A	-	-	-
*V. xuii*	2	Y	negative	+	+	-	-	+	+	A/A	-	-	-
*V. harveyi*	2	Y	negative	+	+	-	-	+	+	A/A	-	-	+
*V. natriegens*	5	Y	negative	+	+	-	-	-	+	A/A	-	-	-
*V. hepatarius*	1	Y	negative	+	+	-	-	+	+	K/A	-	-	-
*P. damselae*	7	G	negative	+	+	+	-	-	+	A/A	-	-	-

**Table 4 microorganisms-10-01851-t004:** The percentage of antibiotic resistance profiles of *Vibrio* spp. collected from seven shrimp farms in different geographical regions of Peninsular Malaysia.

*Vibrio* spp.	No of Isolates	Percentage of Resistance Isolates (%)															
		TET	SXT	E	AMP	C	CN	NOR	K	FEP	CTX	CAZ	KF	CIP	F	VA	P
*V. parahaemolyticus*	124	1	14	19	85	9	7	1	40	2	73	24	86	6	53	97	97
*V. campbellii*	18	6	0	44	100	0	0	0	6	0	50	0	56	11	0	50	56
*V. rotiferianus*	12	50	17	33	83	0	0	0	8	25	50	33	25	0	0	67	83
*V. owensii*	15	0	0	7	100	0	0	0	80	40	80	13	73	27	33	100	87
*V. alginolyticus*	7	14	0	14	86	0	14	0	57	0	57	14	100	0	43	100	100
*V. xuii*	2	0	0	0	50	0	0	0	0	0	0	0	50	0	0	100	100
*V. harveyi*	2	0	0	0	100	0	0	0	0	0	50	0	50	0	50	100	100
*V. natriegens*	5	0	20	20	0	0	20	0	80	0	100	100	100	0	100	100	100
*V. hepatarius*	1	0	0	0	100	0	0	0	100	0	100	0	100	0	100	100	100
*V. communis*	20	5	0	10	95	5	5	0	35	0	70	0	70	5	5	90	90
*V. brasiliensis*	4	0	0	25	75	0	0	0	50	0	0	0	100	0	0	100	100
*Vibrio* spp.	8	0	0	100	100	0	0	0	0	0	25	0	13	13	0	100	100
*P. damselae*	7	0	0	43	100	0	0	0	57	0	0	0	0	0	0	86	71

Abbreviation: AMP: Ampicillin; CTX: Cefotaxime; VA: Vancomycin; P: Penicillin G; KF: Cephalothin; K: Kanamycin; CAZ: Ceftazidime; F: Nitrofurantoin; CN: Gentamicin; E: Erythromycin; SXT: Sulfomethiozole-trimethoprim; FEP: Cefepime; TET: Tetracycline; C: Chloramphenicol; NOR: Norfloxacin; CIP: Ciprofloxacin.

**Table 5 microorganisms-10-01851-t005:** The overall of MAR index of *Vibrio* spp. collected from seven shrimp farms in different geographical regions of Peninsular Malaysia.

AR Index Value	Group of MAR Index	Number of Resistant Antibiotics	Percentage of Isolates (%)
0.06	<0.2	1	0.4
0.13	<0.2	2	4
0.19	<0.2	3	12
0.25	>0.2	4	15
0.31	>0.2	5	16
0.38	>0.2	6	22
0.44	>0.2	7	13
0.50	>0.2	8	6
0.56	>0.2	9	6
0.63	>0.2	10	2
0.69	>0.2	11	3
0.75	>0.2	12	0.4

Abbreviation: MAR: Multiple antibiotic resistance.

**Table 6 microorganisms-10-01851-t006:** Antibiograms and the presence of plasmid of different *Vibrio* spp.

Strain ID	Species	Antibiograms	Presence of Plasmid	No of Resistance Antibiotics
V22G1	*V. parahaemolyticus*	AMP, C, CN, K, CTX, CAZ, KF, F, SXT, E, VA, P	+	12
V22Y2	*V. parahaemolyticus*	AMP, C, CN, K, CTX, CAZ, KF, F, SXT, VA, P	+	11
V5G	*V. parahaemolyticus*	AMP, K, CTX, CAZ, KF, CN, F, SXT, E, VA, P	+	11
V7Y	*V. parahaemolyticus*	AMP, CN, NOR, CTX, CAZ, KF, CIP, F, E, VA, P	+	11
V22Y1	*V. parahaemolyticus*	AMP, C, K, CTX, CAZ, KF, F, SXT, E, VA, P	+	11
V23Y2	*V. parahaemolyticus*	AMP, C, K, CTX, CAZ, KF, F, SXT, E, VA, P	+	11
T26G	*V.parahaemolyticus*	AMP, K, CTX, CAZ, KF, CIP, F, TET, SXT, VA, P	+	11
V3G2	*V. parahaemolyticus*	AMP, K, CTX, CAZ, KF, F, E, C, VA, P	+	10
V4G1	*V. parahaemolyticus*	AMP, K, CTX, CAZ, KF, CIP, CN, F, VA, P	+	10
V7G1	*V. parahaemolyticus*	AMP, K, CTX, CAZ, KF, CIP, F, E, VA, P	+	10
V8G1	*V. parahaemolyticus*	AMP, CN, K, CTX, CAZ, KF, F, SXT, VA, P	+	10
V5Y1	*V. parahaemolyticus*	AMP, C, K, CTX, CAZ, KF, F, SXT, VA, P	+	10
V7G3	*V. parahaemolyticus*	AMP, C, K, CTX, CAZ, KF, F, SXT, VA, P	+	10
V15G	*V. parahaemolyticus*	AMP, C, K, CTX, CAZ, KF, F, SXT, VA, P	+	10
V23G1	*V. parahaemolyticus*	AMP, C, K, CTX, K, F, SXT, E, VA, P	-	10
P10G	*V.parahaemolyticus*	AMP, K, CTX, CAZ, KF, F, SXT, VA, P	+	9
T31G	*V.parahaemolyticus*	AMP, K, CTX, CAZ, KF, F, FEP, VA, P	+	9
V2G	*V. parahaemolyticus*	AMP, K, CTX, CAZ, KF, F, SXT, VA, P	+	9
V3G3-1	*V. parahaemolyticus*	AMP, K, CTX, CAZ, KF, F, E, VA, P	+	9
V4G2	*V. parahaemolyticus*	AMP, K, CTX, CAZ, KF, CIP, F, VA, P	+	9
M5Y1	*V. parahaemolyticus*	AMP, K, CTX, KF, F, E, CN, VA, P	-	9
M5G2	*V. parahaemolyticus*	AMP, CTX, KF, CIP, F, SXT, C, VA, P	+	9
V7G3-1	*V. parahaemolyticus*	AMP, K, CTX, CAZ, KF, F, E, VA, P	+	9
T23G2	*V.parahaemolyticus*	AMP, K, CTX, CAZ, KF, F, VA, P	+	8
T38G1	*V.parahaemolyticus*	AMP, K, CTX, KF, CIP, F, VA, P	+	8
M5Y2	*V. parahaemolyticus*	AMP, K, CTX, KF, F, CN, VA, P	-	8
V1G3	*V. parahaemolyticus*	AMP, K CTX, CAZ, KF, F, VA, P	+	8
V7G2	*V. parahaemolyticus*	AMP, K, CTX, CAZ, KF, F, VA, P	+	8
T39G1	*V.parahaemolyticus*	AMP, K, CTX, CAZ, KF, F, VA, P	+	8
MG2	*V. parahaemolyticus*	AMP, C, CTX, KF, SXT, E, VA, P	-	8
M12-3G	*V. parahaemolyticus*	AMP, K, CTX, KF, F, VA, P	+	7
J11Y	*V. parahaemolyticus*	AMP, K, CTX, KF, F, VA, P	-	7
T1G	*V. parahaemolyticus*	AMP, K, CTX, KF, F, VA, P	+	7
T2G	*V. parahaemolyticus*	AMP, K, CTX, KF, F, VA, P	+	7
T3G	*V. parahaemolyticus*	AMP, K, CTX, KF, F, VA, P	+	7
T7G1	*V. parahaemolyticus*	AMP, K, CTX, KF, F, VA, P	+	7
T11G	*V. parahaemolyticus*	AMP, K, CTX, KF, F, VA, P	+	7
T12Y	*V. parahaemolyticus*	AMP, K, CTX, KF, F, VA, P	+	7
T15G	*V. parahaemolyticus*	AMP, K, CTX, KF, F, VA, P	+	7
T22	*V. parahaemolyticus*	AMP, K, CTX, KF, F, VA, P	+	7
T24G	*V. parahaemolyticus*	AMP, K, CTX, KF, F, VA, P	+	7
V6G	*V. parahaemolyticus*	AMP, K, CTX, CAZ, KF, VA, P	+	7
V22G2	*V. parahaemolyticus*	AMP, CTX CAZ, KF, E, VA, P	+	7
V23Y1	*V. parahaemolyticus*	AMP, K, CTX, KF, F, VA, P	+	7
V3Y3	*V. parahaemolyticus*	AMP, K, KF, F, E, VA, P	+	7
V3Y2G	*V. parahaemolyticus*	AMP, K, KF, F, E, VA, P	+	7
V3Y3G	*V. parahaemolyticus*	AMP, K, KF, F, E, VA, P	+	7
T10Y1	*V. parahaemolyticus*	AMP, K, KF, F, VA, P	+	6
T18G1	*V. parahaemolyticus*	AMP, K, CTX, CAZ, VA, P	+	6
T30G1	*V. parahaemolyticus*	AMP, K, CTX, F, VA, P	+	6
T32G	*V. parahaemolyticus*	AMP, K, CTX, KF, VA, P	+	6
T14G	*V. parahaemolyticus*	AMP, K, CTX, KF, VA, P	+	6
T25G	*V. parahaemolyticus*	AMP, K, CTX, KF, VA, P	+	6
M6-3	*V. parahaemolyticus*	AMP, K, CTX, KF, VA, P	-	6
M6-4	*V. parahaemolyticus*	AMP, K, CTX, KF, VA, P	-	6
M7-3Y	*V. parahaemolyticus*	AMP, K, CTX, KF, VA, P	-	6
V24G1	*V. parahaemolyticus*	AMP, K, KF, E, VA, P	+	6
V1G	*V. parahaemolyticus*	AMP, CTX, F, E, VA, P	+	6
V1G2	*V. parahaemolyticus*	AMP, CTX, KF, F, VA, P	+	6
T21Y2	*V. parahaemolyticus*	AMP, CTX, FEP, F, VA, P	+	6
V4Y3	*V. parahaemolyticus*	AMP, CTX, KF, E, VA, P	+	6
P14G	*V. parahaemolyticus*	AMP, CTX, CAZ, KF, VA, P	-	6
M12-3Y	*V. parahaemolyticus*	AMP, CTX, KF, F, VA, P	+	6
J26Y	*V. parahaemolyticus*	AMP, CTX, KF, F, VA, P	+	6
T4G1	*V. parahaemolyticus*	AMP, CTX, KF, F, VA, P	+	6
T28G	*V. parahaemolyticus*	AMP, CTX, KF, F, VA, P	+	6
T29G1	*V. parahaemolyticus*	AMP, CTX, KF, F, VA, P	+	6
M11-4Y	*V. parahaemolyticus*	AMP, CAZ, KF, FEP, VA, P	+	6
J29Y2	*V. parahaemolyticus*	K, CTX, KF, F, VA, P	-	6
J21Y	*V. parahaemolyticus*	K, CTX, KF, F, VA, P	-	6
J31Y	*V. parahaemolyticus*	K, CTX, KF, CIP, VA, P	-	6
J29Y1	*V. parahaemolyticus*	CTX, CAZ, KF, F, VA, P	-	6
V14G2	*V. parahaemolyticus*	AMP, K, CTX, F, VA, P	+	6
MG2Y	*V. parahaemolyticus*	AMP, K, CTX, F, VA, P	+	6
M2Y2-1	*V. parahaemolyticus*	AMP, K, CTX, KF, VA, P	-	6
V14G1	*V. parahaemolyticus*	AMP, CTX, KF, SXT, VA, P	+	6
V25G	*V. parahaemolyticus*	AMP, CTX, KF, E, VA, P	+	6
V3G1	*V. parahaemolyticus*	AMP, KF, E, VA, P	+	5
V3Y2Y	*V. parahaemolyticus*	AMP, KF, E, VA, P	+	5
V24G2	*V. parahaemolyticus*	AMP, KF, E, VA, P	+	5
P33G	*V. parahaemolyticus*	AMP, K, KF, VA, P	-	5
T19G	*V. parahaemolyticus*	AMP, K, KF, VA, P	+	5
M4-4G	*V. parahaemolyticus*	AMP, K, CTX, VA, P	+	5
T33	*V. parahaemolyticus*	AMP, CTX, F, VA, P	+	5
M9-4Y	*V. parahaemolyticus*	AMP, CTX, KF, VA, P	-	5
M10-3Y	*V. parahaemolyticus*	AMP, CTX, KF, VA, P	+	5
M10-4	*V. parahaemolyticus*	AMP, CTX, KF, VA, P	-	5
M11-3Y	*V. parahaemolyticus*	AMP, CTX, KF, VA, P	-	5
M11-4G	*V. parahaemolyticus*	AMP, CTX, KF, VA, P	-	5
T8Y2	*V. parahaemolyticus*	AMP, CTX, KF, VA, P	+	5
T13G	*V. parahaemolyticus*	AMP, CTX, KF, VA, P	+	5
T16G	*V. parahaemolyticus*	AMP, CTX, KF, VA, P	+	5
J1Y	*V. parahaemolyticus*	K, CTX, KF, VA, P	-	5
J10Y	*V. parahaemolyticus*	K, CTX, KF, VA, P	-	5
J17Y1	*V. parahaemolyticus*	CTX, KF, F, VA, P	-	5
J23	*V. parahaemolyticus*	CTX, KF, F, VA, P	-	5
P21G	*V. parahaemolyticus*	AMP, E, VA, P	+	4
T6G	*V. parahaemolyticus*	AMP, F, VA, P	+	4
M4-3Y	*V. parahaemolyticus*	AMP, K, VA, P	+	4
J14Y	*V. parahaemolyticus*	AMP, KF, VA, P	-	4
J15Y	*V. parahaemolyticus*	AMP, KF, VA, P	-	4
M9-3G	*V. parahaemolyticus*	AMP, KF, VA, P	-	4
M9-3Y	*V. parahaemolyticus*	AMP, KF, VA, P	-	4
M9-4G	*V. parahaemolyticus*	AMP, KF, VA, P	-	4
J25Y2	*V. parahaemolyticus*	AMP, KF, VA, P	-	4
P29G	*V. parahaemolyticus*	AMP, KF, VA, P	-	4
M2-4Y	*V. parahaemolyticus*	AMP, K, VA, P	-	4
J3Y2	*V. parahaemolyticus*	KF, F, VA, P	-	4
J6	*V. parahaemolyticus*	KF, F, VA, P	-	4
J17Y2	*V. parahaemolyticus*	KF, F, VA, P	-	4
J18Y	*V. parahaemolyticus*	KF, F, VA, P	-	4
J9Y	*V. parahaemolyticus*	K, KF, VA, P	-	4
J4Y	*V. parahaemolyticus*	CTX, KF, VA, P	-	4
J24Y1	*V. parahaemolyticus*	CTX, KF, VA, P	-	4
S2-4	*V. parahaemolyticus*	AMP, VA, P	+	3
S12-3	*V. parahaemolyticus*	AMP, VA, P	+	3
S15-4	*V. parahaemolyticus*	AMP, VA, P	-	3
S19-3	*V. parahaemolyticus*	AMP, VA, P	+	3
M4-4Y	*V. parahaemolyticus*	AMP, VA, P	-	3
M5-3G	*V. parahaemolyticus*	AMP, VA, P	+	3
P22G	*V. parahaemolyticus*	AMP, VA, P	-	3
J5Y	*V. parahaemolyticus*	KF, VA, P	-	3
J24Y2	*V. parahaemolyticus*	KF, VA, P	-	3
J25Y1	*V. parahaemolyticus*	KF, VA, P	-	3
K36Y1	*V. communis*	AMP, K, CTX, KF, F, VA, P	+	7
K8Y1	*V. communis*	AMP, K, CTX, KF, VA, P	-	6
K14Y1	*V. communis*	AMP, K, CTX, KF, VA, P	-	6
K18Y1	*V. communis*	AMP, K, CTX, KF, VA, P	-	6
K19Y1	*V. communis*	AMP, K, CTX, KF, VA, P	-	6
K22Y	*V. communis*	AMP, K, CTX, KF, VA, P	+	6
K33Y2	*V. communis*	AMP, K, CTX, KF, VA, P	+	6
K28Y1	*V. communis*	AMP, CN, CTX, KF, VA, P	+	6
K9Y	*V. communis*	AMP, CTX, KF, E, VA, P	-	6
P11Y	*V. communis*	AMP, CTX, KF, TET, VA, P	-	6
S1-3	*V. communis*	AMP, CTX, CIP, VA, P	+	5
K4Y1	*V. communis*	AMP, C, CTX, VA, P	-	5
K7Y	*V. communis*	AMP, K, KF, VA, P	+	5
K13Y	*V. communis*	E, CTX, KF, VA, P	-	5
S1-4	*V. communis*	AMP, CTX, VA, P	+	4
K1Y1	*V. communis*	AMP, KF, VA, P	-	4
P13Y2	*V. communis*	AMP, KF, VA, P	-	4
P1Y	*V. communis*	AMP, VA, P	-	3
P23Y2	*V. communis*	AMP, VA, P	-	3
P25G	*V. communis*	AMP, VA, P	+	3
S7-4	*V. campbellii*	AMP, CTX, KF, CIP, VA, P	+	6
S23-4	*V. campbellii*	AMP, CTX, KF, CIP, VA, P	+	6
S10-4	*V. campbellii*	AMP, CTX, KF, VA, P	+	5
S17-3	*V. campbellii*	AMP, CTX, KF, VA, P	+	5
S18-4	*V. campbellii*	AMP, CTX, KF, VA, P	+	5
S31-3	*V. campbellii*	AMP, CTX, KF, VA, P	+	5
S31-4	*V. campbellii*	AMP, CTX, KF, VA, P	+	5
M8-4Y	*V. campbellii*	AMP, CTX, CAZ, VA, P	-	5
P24G	*V. campbelliii*	AMP, CTX, VA, P	-	4
P28G2	*V. campbelliii*	AMP, TET, VA, P,	-	4
P31Y	*V. campbellii*	AMP, KF, VA, P	-	4
S10-3	*V. campbellii*	AMP, KF, VA, P	+	4
P21Y	*V. campbellii*	AMP, VA, P	-	3
P25Y1	*V. campbellii*	AMP, VA, P	-	3
P32Y	*V. campbellii*	AMP, VA, P	-	3
P8Y	*V. campbellii*	AMP, K, P	-	3
P25Y2	*V. campbellii*	AMP, P	-	2
P3Y1	*V. campbellii*	AMP, P	-	2
T17G4	*V. owensii*	AMP, FEP, CTX, CAZ, KF, K, CIP, VA, P	+	9
T34Y	*V. owensii*	AMP, FEP, CTX, CAZ, KF, F, E, VA, P	+	9
T20G2	*V. owensii*	AMP, K, FEP, CTX, KF, CIP, VA, P	+	8
T23G1	*V. owensii*	AMP, K, FEP, CTX, KF, CIP, VA, P	+	8
T23G5	*V. owensii*	AMP, K, FEP, CTX, CIP, VA, P	+	7
K16Y2	*V. owensii*	AMP, K, CTX, KF, F, VA, P	-	7
K38Y2	*V. owensii*	AMP, K, CTX, KF, F, VA, P	+	7
M6Y	*V. owensii*	AMP, K, CTX, KF, F, VA, P	+	7
T5Y1	*V. owensii*	AMP, FEP, CTX, KF, VA, P	+	7
K3Y2	*V. owensii*	AMP, K, CTX, KF, VA, P	-	6
K16Y1	*V. owensii*	AMP, K, CTX, KF, VA, P	-	6
K21Y	*V. owensii*	AMP, K, CTX, KF, VA, P	+	6
K38Y1	*V. owensii*	AMP, CTX, F, E, VA, P	-	6
P27G	*V. owensii*	AMP, KF, VA, P	-	4
P34Y	*V. owensii*	AMP, VA, P	-	3
K34Y2	*V. rotiferianus*	AMP, FEP, CTX, CAZ, TET, SXT, E, VA, P	+	9
K37Y1	*V. rotiferianus*	AMP, FEP, CTX, KF, F, TET, E, VA, P	+	9
P20Y1	*V. rotiferianus*	AMP, FEP, CTX, CAZ, KF, TET, VA, P	-	8
K26Y	*V. rotiferianus*	AMP, CTX, CAZ, TET, SXT, E, VA, P	+	8
M5-3Y	*V. rotiferianus*	AMP, K, CTX, KF, F, VA, P	-	7
S15-3	*V. rotiferianus*	AMP, K, KF, E, VA, P	+	6
K17Y1	*V. rotiferianus*	AMP, CTX, TET, VA, P	-	5
S24-4	*V. rotiferianus*	AMP, VA, P	-	3
K30Y	*V. rotiferianus*	AMP, VA, P	+	3
P12Y	*V. rotiferianus*	AMP, CTX, P	-	3
K17Y2	*V. rotiferianus*	TET, VA, P	-	3
P7G	*V. rotiferianus*	P	-	1
V1Y	*V. natriegens*	E, CN, K, CTX, CAZ, KF, F, VA, P	-	9
V1G2Y	*V. natriegens*	SXT, CTX, CAZ, KF, F, VA, P	+	7
V1Y2	*V. natriegens*	K, CTX, CAZ, KF, F, VA, P	+	7
V2Y	*V. natriegens*	K, CTX, CAZ, KF, F, VA, P	-	7
V4Y1	*V. natriegens*	K, CTX, CAZ, KF, F, VA, P	-	7
K8Y2	*V.brasiliensis*	AMP, K, KF, VA, P	-	5
K40Y	*V. brasiliensis*	AMP, KF, E, VA, P	+	5
K6Y	*V. brasiliensis*	AMP, KF, VA, P	+	4
K24Y	*V. brasiliensis*	K, KF, VA, P	-	4
P19Y	*V. harveyi*	AMP, CTX, KF, F, VA, P	-	6
P23Y1	*V. harveyi*	AMP, VA, P	-	3
P29Y	*V. xuii*	AMP, KF, VA, P	-	4
P13Y1	*V. xuii*	KF, VA, P	-	3
K37Y2	*V. alginolyticus*	AMP, CN, K, CTX, KF, F, TET, VA, P	+	9
K27Y1	*V. alginolyticus*	AMP, K, CTX, CAZ, KF, VA, P	+	7
K36Y2	*V. alginolyticus*	AMP, K, KF, F, E, VA, P	-	7
K5Y	*V. alginolyticus*	AMP, K, CTX, KF, VA, P	+	6
K41Y	*V. alginolyticus*	AMP, CTX, KF, VA, P	+	5
J3Y1	*V. alginolyticus*	KF, F, VA, P	-	4
K27Y2	*V. alginolyticus*	AMP, VA, P	+	3
M13Y	*V. hepatarius*	AMP, K, CTX, KF, F, VA, P	-	7
S6-3	*Vibrio *spp.	AMP, KF, CIP, VA, P	+	5
S3-4	*Vibrio *spp.	CTX, VA, P	-	3
S3-3	*Vibrio *spp.	VA, P	-	2
S5-4	*Vibrio *spp.	VA, P	-	2
S21-3	*Vibrio *spp.	VA, P	+	2
S25-3	*Vibrio *spp.	VA, P	+	2
S34-4	*Vibrio *spp.	VA, P	-	2
S35-4	*Vibrio *spp.	VA, P	-	2
M14-4Y	*P. damselae*	AMP, CN, CTX, KF, VA, P	-	6
M8-4G	*P. damselae*	AMP, K, CTX, KF, VA, P	-	6
S27-3	*P. damselea*	AMP, K, VA, P	+	4
M2-4G	*P. damselae*	AMP, K, VA, P	+	4
M1-4Y	*P. damselae*	AMP, E, VA, P	-	4
M2-3Y	*P. damselae*	AMP, E, VA, P	-	4
M4Y1G	*P. damselea*	AMP, VA	+	2

Abbreviation: AMP: Ampicillin; CTX: Cefotaxime; VA: Vancomycin; P: Penicillin G; KF: Cephalothin; K: Kanamycin; CAZ: Ceftazidime; F: Nitrofurantoin; CN: Gentamicin; E: Erythromycin; SXT: Sulfomethiozole-trimethoprim; FEP: Cefepime; TET: Tetracycline; C: Chloramphenicol; NOR: Norfloxacin; CIP: Ciprofloxacin; +: Isolates that harbored plasmid; -: Isolates that did not harbored plasmid.

**Table 7 microorganisms-10-01851-t007:** Plasmid profiles of the resistance *Vibrio* isolates (n = 125) in this study.

No of Plasmid	Plasmid Profiles	Plasmid Size (kb)	No of Isolates (%)
1	1	1.3	4 (3.2)
	2	3.0	1 (0.8)
	3	5.0	2 (1.6)
	4	above 10.0	65 (52.0)
2	5	1.0, 3.0	1 (0.8)
	6	1.0, above 10.0	2 (1.6)
	7	1.5, 3.0	1 (0.8)
	8	2.5, above 10.0	2 (1.6)
	9	3.0, 5.0	1 (0.8)
	10	3.0, above 10.0	1 (0.8)
	11	4.0, above 10.0	1 (0.8)
	12	5.0, above 10.0	9 (7.2)
	13	6.0, above 10.0	1 (0.8)
3	14	1.3, 3.0, above 10.0	2 (1.6)
	15	1.5, 2.5, above 10.0	1 (0.8)
	16	2.0, 5.0, above 10.0	2 (1.6)
	17	2.5, 5.0, above10.0	5 (4.0)
	18	3.0, 5.0, above10.0	13 (10.4)
4	19	1.0, 3.0, 5.0, above 10.0	2 (1.6)
	20	1.5, 3.0, 5.0, above 10.0	2 (1.6)
5	21	1.0, 1.5, 2.0, 3.0, above 10.0	1 (0.8)
	22	3.0, 4.0, 5.0, 6.0, above 10.0	2 (1.6)
6	23	1.0, 1.5, 2.0, 3.0, 5.0, above 10.0	4 (3.2)
		**Total**	**125 (100)**

## Data Availability

Not applicable.

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
