# Peer review of "Prevalence, Antibiotics Resistance and Plasmid Profiling of Vibrio spp. Isolated from Cultured Shrimp in Peninsular Malaysia"

_microorganisms, 2022, doi:10.3390/microorganisms10091851_

Round 1
Reviewer 1 Report
Great work on this manuscript and study. I think the profiling of Vibro sp. plasmids and species is important and relevant work, particularly as more antibiotic resistant microbes ride worldwide. It is novel and interesting to examine this in shrimp as well, and to analyze the plasmids in particular. And I personally love the rejection of 16S for species ID and using other genes as differentiators. I think MLST - or using several genes, would have been nice and I would love to see a comparison between those genes and 16S to drive home that point - although that is a bit off-topic I realize and seems to have been well addressed by others, as explained in the discussion.
For the criticisms: figure 4 and figure 2 seem quite a bit redundant - I'm not sure what figure 4 really adds as it's clear from figure 2 and 3 what the breakdown in vibrio species is. The table also shows this information as well - so figure 4 could go if needed. Instead it could be nice to show your antibiotic resistance gene profiling overlayed on the species breakdown. Are more V. parahemolyticus resistant to more antibiotics than others? You have all of this data and I think that's a relevant question which antibiotic resistance gene or SNP goes to which species. I'd also like to see some discussion of the different species of Vibrio and which are more pathogenic to shrimp and to humans to link together the relevance.
Figure 5: I also am left wondering which species of vibrio has these plasmids and antibiotic resistances - I think data visualization could be greatly improved and you could combine tables and graphs and probably put some of these tables into supplement. I'm not sure its very important to show the size of the plasmids, but rather do some work to show what antibiotic resistance (s) is ON the plasmids.
Author Response
Dear editor and reviewer,
Greetings!
Thank you for your opportunity to revise our paper titled “Prevalence, antibiotics susceptibility and their plasmid profiling of Vibrio spp. isolated from cultured shrimps in Peninsular Malaysia”. On behalf of all the authors, there is no conflict of interest about the study submitted to the journal for possible publication.
Corrections and rebuttals have been made in response to your email regarding the review of our manuscript as follows:
Response to reviewer 1 comments
Point 1: Great work on this manuscript and study. I think the profiling of Vibrio sp. plasmids and species is important and relevant work, particularly as more antibiotic resistant microbes ride worldwide. It is novel and interesting to examine this in shrimp as well, and to analyze the plasmids in particular. And I personally love the rejection of 16S for species ID and using other genes as differentiators.
Response 1: Thank you for the comments. Authors really appreciated them.
Point 2: I think MLST - or using several genes, would have been nice and I would love to see a comparison between those genes and 16S to drive home that point - although that is a bit off-topic I realize and seems to have been well addressed by others, as explained in the discussion.
Response 2: Thank for the suggestion. Authors really appreciated them. Authors will plan do it for the next manuscript.
Point 3: For the criticisms: figure 4 and figure 2 seem quite a bit redundant - I'm not sure what figure 4 really adds as it's clear from figure 2 and 3 what the breakdown in vibrio species is. The table also shows this information as well - so figure 4 could go if needed.
Response 3: Thank you for the comments. After discussion with other authors, we decided to remove Figure 4 from the manuscript to avoid redundancy and confusing the readers.
Point 4: Instead it could be nice to show your antibiotic resistance gene profiling overlayed on the species breakdown. Are more V. parahemolyticus resistant to more antibiotics than others? You have all of this data and I think that's a relevant question which antibiotic resistance gene or SNP goes to which species
Response 4: Thank you for the suggestion. Authors have made necessary changes in the related matter according to the suggestion. The changes were in red. Kindly refer to the:
- Line 326-331; 367-375; Results: 3.6. Antibiotics susceptibility
- Table 4
- Line 609-616 for additional discussion
Point 5: I'd also like to see some discussion of the different species of Vibrio and which are more pathogenic to shrimp and to humans to link together the relevance
Response 5: Thank you for the suggestion. Authors have made necessary changes in the discussion according to the suggestion as follows. The changes were made in line 520-531 in red.
Vibrio spp. are a group of bacteria naturally found in freshwater, estuarine and marine environments [38]. It is commonly known that Vibrio spp. are responsible for numerous of human diseases attributed to the natural microbiota of aquatic environments and seafood [39]. In humans, Vibrio spp. are known to cause gastroenteritis, cholera, and septicemia [40]. The human pathogenic Vibrio spp. of clinical relevance including V. parahaemolyticus, V. alginolyticus, V. cholerae, V. vulnificus, V. tubiashi, and V. fluvialis [41]. In addition, V. parahaemolyticus, V. vulnificus, and V. mimicus are foodborne pathogens [42]. Meanwhile, the common Vibrio spp. associated with the shrimp diseases were V. harveyi, V. parahaemolyticus, V. alginolyticus, V. anguillarum, V. vulnificus and V. splendidus. V. harveyi is associated with luminescent vibriosis in shrimps particularly in P. vannamei and P. monodon. While V. parahaemolyticus can cause human illness and is frequently associated with food-borne gastroenteritis or diarrhoea [43].
Point 6: Figure 5: I also am left wondering which species of vibrio has these plasmids and antibiotic resistances - I think data visualization could be greatly improved and you could combine tables and graphs and probably put some of these tables into supplement. I'm not sure its very important to show the size of the plasmids, but rather do some work to show what antibiotic resistance (s) is ON the plasmids.
Response 6: Thank you for the suggestion. Authors have made improvement according to the suggestion. Figure 5 have removed and replaced with Table 6. The changes were in red.
Kindly refer to:
- Line 400 - 408; Result; 3.8. Plasmid profiling
- Table 6
Please see the attachment on the manuscript. Thank you.
Kindly regards,
Haifa-Haryani

Reviewer 2 Report
My decision is “minor revision”
Author Response
Dear editor and reviewer,
Greetings!
Thank you for your opportunity to revise our paper titled “Prevalence, antibiotics susceptibility and their plasmid profiling of Vibrio spp. isolated from cultured shrimps in Peninsular Malaysia”. On behalf of all the authors, there is no conflict of interest about the study submitted to the journal for possible publication.
Corrections and rebuttals have been made in response to your email regarding the review of our manuscript as follows:
Response to reviewer 2 comments
Point1: The study entitled “Prevalence, antibiotics susceptibility and their plasmid profiling of Vibrio spp. isolated from cultured shrimps in Peninsular Malaysia” provide some important data regarding antibiotic resistance of Vibrio spp. in farmed shrimp, however there are certain issues that need more explanation.
Response 1: Thank you for the comments. Authors will try their best to improve the manuscript accordingly.
Point 2: Correct the title or choose more descriptive title
Eg: Antibiotics resistance of Vibrio spp. isolated from cultured shrimp in Peninsular Malaysia and their plasmid profiling.
Response 2: Thank you for the suggestion. Authors have made necessary changes regarding the title. We agreed to the suggested title. However, after discussion with the other authors, we decided to remain the words “prevalence” in the title since we would like the see the distribution of Vibrio spp. from Peninsular Malaysia.
The new title for the manuscript: Prevalence, antibiotics resistance and their plasmid profiling of Vibrio spp. isolated from cultured shrimp in Peninsular Malaysia.
Point 3: Abstract
line 28 … aimed
Response 3: Thank you for the comment. Authors have replaced “aims” to “aimed”
Point 4: Abstract
Shrimps…………….shrimp
Response 4: Thank you for the comment. Authors have replaced “shrimps” to “shrimp”
Point 5: Introduction
Write short paragraph in the introduction about alternatives to antibiotics that may be used to control infectious diseases affecting farmed shrimp.
Response 5: Thank you for the suggestion. Authors have made necessary changes in the introduction according to the suggestion as follows. The changes were in red.
Line 69-76
Several alternatives were considered to control the outbreak in shrimp farms, including strict biosecurity measures, green water systems, probiotics, and phage [12,13]. In fact, a better farm management including seed and stock selection, aeration, water treatment, and the application of non-harmful chemicals such as organic acids and natural products in the diet has been considered [14,15,16]. A good feed management system, including high-quality feed enriched with immunostimulants, as well as the addition of probiotics, will aid in disease prevention [17]. Thus, ignoring the biosecurity of shrimp hatcheries and farms allows the outbreak to spread.
Point 6: Material and methods
-How you collected the shrimp samples from the studied farms in a random manner?
Response 6: Thank you for the comment. Authors have made necessary changes in the material and methods according to the suggestion as follows. The changes were in red.
Line 92 – 101:
A total of 210 shrimp were randomly collected between March 2019 and March 2021 from seven farms (n=30 shrimp/farm) that were located around Peninsular Malaysia (Table 1). They were P. monodon (n = 150) and P. vannamei (n = 60) of 30 days of culture. Shrimp sample including healthy and unhealthy shrimp were randomly collected by using lift net. During the collection, shrimp length and weight were measured and recorded. Then, the samples were immediately placed in the ice and transferred to the laboratory within 3 to 4 h. Samples were washed thoroughly with sterile distilled water followed by dissection with sterile scissors. Following collection, the hepatopancreas and midgut were carefully examined. Healthy shrimp showed large and black hepatopancreas with a full midgut, whereas unhealthy shrimp had small and pale hepatopancreas with an empty midgut. Other clinical sign such as body coloration, growth, size of the shrimp, sign of the poor feeding and swimming behaviour were observed [22,23,24].
Point 7: Material and methods
-What the other clinical signs noticed in the affected shrimp other than that mentioned (small and pale hepatopancreas with an empty midgut)
Response 7: Thank you for the comment. Authors have made necessary changes in the material and methods according to the suggestion as follow. The changes were in red.
Line 100 – 101:
A total of 210 shrimp were collected between March 2019 and March 2021 from seven farms (n=30 shrimp/farm) that were located around Peninsular Malaysia (Table 1). They were P. monodon (n = 150) and P. vannamei (n = 60) of 30 days of culture. Shrimp sample including healthy and unhealthy shrimp were randomly collected by using lift net. During the collection, shrimp length and weight were measured and recorded. Then, the samples were immediately placed in the ice and transferred to the laboratory within 3 to 4 h. Samples were washed thoroughly with sterile distilled water followed by dissection with sterile scissors. Following collection, the hepatopancreas and midgut were carefully examined. Healthy shrimp showed large and black hepatopancreas with a full midgut, whereas unhealthy shrimp had small and pale hepatopancreas with an empty midgut. Other clinical sign such as body coloration, growth, size of the shrimp, sign of the poor feeding and swimming behaviour were observed [22,23,24].
Point 8: Material and methods
- Thiosulfate citrate bile salts sucrose (TCBS)
Response 8: Thank you for the comment. Authors have changed thiosulfate citrate bile salts to thiosulfate citrate bile salts sucrose (TCBS) (Line 110).
Point 9: Material and methods
- Are all the bacterial colonies grown onto (TCBS medium) identified as Vibrio spp.? Had you detected any false positive Vibrio spp. onto TCBS?
Response 9: Thank you for the comment. Commonly known that, TCBS is widely used in the isolation of vibrios in many different types of samples.The was no false positive control were detected in this study. In fact, we performed biochemical test for each sample and confirmed the each of the samples by molecular identification using pyrH gene. The findings shows that all the isolates produced a single band of 500 bp indicating that the samples were Vibrio spp.
Point 10: Material and methods
-How you performed the biochemical identification of Vibrio spp.?
Response 10: Thank you for the comment. In this study, seven type of biochemical identification of Vibrio spp. were performed including Gram staining, oxidase, catalase, ONPG, SIM, TSI and LDC. The biochemical identification was performed according to Jayasinge et al., 2008 and Buller, 2015.
The method was included in the manuscript in line 114 – 119 (2.3.Biochemical tests).
Point 11: Results
-line 199 replace with “clinical examination
Response 11: Thank you for the comment. Authors have replaced with clinical examination.
Point 12: Results
-mention the other clinical signs noticed in the infected shrimp
Response 12: Thank you for the comment. Authors have made necessary changes in the result according to the suggestion as follows. The changes were in red. Line 203 - 205
Most of the sampled shrimp were healthy (85%). While 15% were unhealthy shrimp. Based on the observation, hepatopancreas of healthy shrimp was large and black, while the midgut was full (Figure 1a, Figure 1c). On the other hand, hepatopancreas of unhealthy shrimp appeared small and pale, while the midgut was empty (Figure 1b, Figure 1d). The body coloration between healthy and unhealthy seems to be different which was unhealthy shrimp showed a pale body colour (Figure 1b) compared to healthy shrimp (Figure 1a). In fact, the size of unhealthy shrimp was smaller compared to healthy shrimp. The in-situ examination showed that the hepatopancreas and midgut were associated with the sign of vibriosis. However, additional test in terms of molecular identification and histology examination are required for the verification.
Point 13: Results
- What are the other detected bacterial spp. than Vibrio spp.?
Response 13: Thank you for the comment. In this study, there was no other detected bacterial spp. than Vibrio spp. During isolation, green and yellow colony were picked from TCBS. Then, biochemical test was performed, and the findings showed that the isolates were considered as a presumptive Vibrio spp. Then, molecular detection using pyrH was performed as a confirmation. The findings shows that all the isolates produced a single band of 500 bp indicating that the samples were Vibrio spp.
Point 14: Results
- Had you detected any false positive Vibrio spp. onto the TCBS medium?
Response 14: Thank you for the comment. There was no false positive Vibrio spp. onto the TCBS medium were detected since biochemical test and molecular identification were performed. The findings found that the isolates were positive for the Vibrio spp.
Point 15: Results
-Which Vibrio spp. produced yellow colonies and that produced green colonies. Add this information in your results in table 3.
Response 15: Thank you for the suggestion. Authors have made necessary changes according to the suggestion. New results have been added in Table 3. The changes were in red. Kindly refer line 276
Point 16: Results
- Clarify the mixed infections recorded in your study
Response 16: Thank you for the comment. The shrimp were randomly collected from the selected farms. Furthermore, the shrimp were observed for the sign of vibriosis during in-situ examination. Hence, no mixed infection was recorded in this study.
Point 17: Results
- How could you explain existence of resistance after plasmid curing?
Response 17: Thank you for the comment. Authors have made necessary changes in the result according to the suggestion as follows. The changes were in red.
Line 463 - 468
….For norfloxacin (NOR), the isolates were changed from resistance to susceptible after the curing process (Figure 4a, Figure 4b). There was no change in resistance to sulfomethiozole-trimethoprim (SXT) and tetracycline (TET) after plasmid curing, indicating that the resistance isolates were chromosomal mediated (Figure 4a). The result after plasmid curing revealed that when the antibiotic resistance profile is affected, it indicates that the resistance isolates is plasmid-mediated, whereas when the antibiotic resistance profile is unaffected, it indicates that the resistance isolates is chromosomal-mediated [19].
Point 18: Results
- Are plasmids were not detected in any of the non-antibiotic resistant strains?
Response 18: Thank you for the comment. Authors have made necessary changes in the result according to the suggestion as follows. Th changes were in red. Kindly refer to:
- Result 3.8.Plasmid profiling ; Line 401 - 408
- Table 6; Line 420
- Line 643 – 651 (Discussion)
Point 19: Could you establish a statistical correlation between the phenotypic antibiotic resistance of isolates (antibiogram profile) noticed in your study with the agar diffusion method and the detection of plasmid in Vibrio isolates?
Response 19: Thank you for the suggestion. Authors really appreciated them. Authors will do the related matter in the next study.
Point 20: Discussion
Discussion section is scanty. The findings of your investigation need more emphasis.
Response 20: Thank you for the comments. Authors will try their best to improve the discussion accordingly. The changes were in red.
Please see the attachment for the revised manuscript
Kindly regards
Haifa-Haryani
